behaviour, ecology, evolution

cooperative breeding, polygynandry, long-term data, reproductive conflict, reproductive costs, reproductive benefits

**Author for correspondence:**
Sahas Barve
e-mail: sahasbarve@gmail.com

# Lifetime reproductive benefits of cooperative polygamy vary for males and females in the acorn woodpecker (*Melanerpes formicivorus*)

Sahas Barve[1], Christina Riehl[2], Eric L. Walters[3], Joseph Haydock[4], Hannah L. Dugdale[5] and Walter D. Koenig[6,7]

[1]Division of Birds, Smithsonian National Museum of Natural History, 10th Street and Constitution Avenue NW, Washington, DC 20560, USA
[2]Department of Ecology and Evolutionary Biology, Princeton University, 106A Guyot Hall, Princeton, NJ 08544, USA
[3]Department of Biological Sciences, Old Dominion University, Norfolk, VA 23529, USA
[4]Department of Biology, Gonzaga University, Spokane, WA 99258, USA
[5]Groningen Institute for Evolutionary Life Sciences, University of Groningen, Groningen 9747 AG, The Netherlands
[6]Hastings Reservation, University of California Berkeley, 38601 E. Carmel Valley Rd., Carmel Valley, CA 93924, USA
[7]Lab of Ornithology, Cornell University, 159 Sapsucker Woods Rd., Ithaca, NY 14850, USA

SB, 0000-0001-5840-8023; CR, 0000-0002-7872-336X; ELW, 0000-0002-9414-5758; HLD, 0000-0001-8769-0099; WDK, 0000-0001-6207-1427

Cooperative breeding strategies lead to short-term direct fitness losses when individuals forfeit or share reproduction. The direct fitness benefits of cooperative strategies are often delayed and difficult to quantify, requiring data on lifetime reproduction. Here, we use a longitudinal dataset to examine the lifetime reproductive success of cooperative polygamy in acorn woodpeckers (*Melanerpes formicivorus*), which nest as lone pairs or share reproduction with same-sex cobreeders. We found that males and females produced fewer young per successful nesting attempt when sharing reproduction. However, males nesting in duos and trios had longer reproductive lifespans, more lifetime nesting attempts and higher lifetime reproductive success than those breeding alone. For females, cobreeding in duos increased reproductive lifespan so the lifetime reproductive success of females nesting in duos was comparable to those nesting alone and higher than those nesting in trios. These results suggest that for male duos and trios, reproductive success alone may provide sufficient fitness benefits to explain the presence of cooperative polygamy, and the benefits of cobreeding as a duo in females are higher than previously assumed. Lifetime individual fitness data are crucial to reveal the full costs and benefits of cooperative polygamy.

## 1. Introduction

Cooperative breeding, in which more than two adults cooperate to raise a single brood of young, is a widespread reproductive strategy in both vertebrates [1] and invertebrates [2]. Most cooperatively breeding vertebrates live in family groups that consist either of a breeding pair and related non-breeding helpers, or of cobreeders that share reproduction of the brood with or without helpers [3]. Empirical studies have consistently found that helpers and cobreeders have lower annual reproductive output than individuals that breed independently: helpers typically fail to reproduce altogether, while cobreeders often compete for reproduction within the social group, leading to lower per capita reproductive success [4]. The repeated evolution of cooperative breeding despite these

fitness costs has fascinated evolutionary biologists for decades [5]. Kin selection provides a powerful explanation for the evolutionary persistence of cooperation among relatives, since the indirect fitness gained from raising non-descendant kin can mitigate the direct fitness costs of cooperation [3,6].

Cooperative strategies such as helping or cobreeding may also yield delayed direct fitness benefits, such as increased survival or reproductive output later in life, which may compensate for the reproductive costs suffered by a cooperator for a given breeding attempt [7]. This delayed fitness hypothesis has been tested in several species in which a breeding pair is assisted by related, non-breeding helpers. In red wolves (*Canis rufus*) [8], red-cockaded woodpeckers (*Dryobates borealis*) [9] and green woodhoopoes (*Phoeniculus purpureus*) [10], for example, offspring that remain as helpers on their natal territory have higher survival than those that disperse, leading to equal (or greater) lifetime reproductive success for offspring that cooperate relative to those that disperse. Thus, in such species, reproductive gains later in life may partly explain the evolution of helping behaviour [11].

Cooperative polygamy differs from helper-at-the-nest societies in several important ways. Although cobreeders gain a direct reproductive share in the nesting attempt, clutch or litter size is typically limited, so individual reproductive output is lower for cobreeders than for single breeders [12]. Whereas non-breeding helpers often remain with their natal group for only 1–2 years before dispersing, alliances between same-sex cobreeders can last for many years, and thus the costs of sharing reproduction in cooperatively polygamous groups may be borne throughout reproductive maturity [13,14]. However, if cobreeding confers other ecological benefits such as obtaining a higher quality territory or longer reproductive tenure, these benefits may balance the short-term costs of shared reproduction, suggesting that cooperative polygamy may be maintained by the lifetime number of young fledged (direct fitness) alone, without the additional benefits accruing to related individuals (indirect fitness). Lifetime measures of reproductive success can reveal the delayed fitness benefits of cooperative polygamy and are thus crucial to understanding the evolution and maintenance of this phenomenon [8,15,16].

We compared the lifetime reproductive success of single breeding versus cooperative polygamy in the acorn woodpecker (*Melanerpes formicivorus*), using a 43-year dataset that quantified the lifetime reproductive output of 499 breeding adults. Acorn woodpeckers in the study population breed as lone pairs or in groups containing multiple breeding males and females; breeders may also be assisted by non-reproductive helpers of either sex (approx. 65% of groups). First, we tested the hypothesis that lifetime reproductive output differed between adults that did not share same-sex reproduction (single breeders) and those that bred as same-sex duos, trios, or even larger coalitions. Second, we examined the effects of cobreeding on several demographic parameters that could contribute to observed differences in lifetime reproductive output, including age at first reproduction, the number of offspring produced per successful breeding attempt, reproductive lifespan (number of years an individual was present in the population as a breeder) and number of lifetime reproductive attempts (number of nests initiated by the group during the individual's tenure as a breeder). Finally, we investigated associations between same-sex cobreeding and ecological factors that could influence demographic parameters, including

territory quality and the outcome of inter-group conflict over breeding vacancies. Our goal was to test whether cooperative polygamy resulted in lower lifetime reproductive success compared with single breeding, or whether the reproductive fitness benefits of these alternative strategies were comparable.

## 2. Methods

### (a) Ecology of acorn woodpeckers

Acorn woodpeckers are sexually dimorphic birds found in oak (*Quercus* spp.)-dominated habitats from the western United States to Colombia [17]. They have been studied at Hastings Natural History Reservation in Carmel Valley, central coastal California, USA (36.387° N, 121.551° W) since 1968 [18]. Woodpeckers in this population live in polygynandrous social groups with or without non-breeding helpers, individuals that remain on their natal territory with their parents and assist in rearing non-descendant offspring [19]. Helping behaviour in acorn woodpeckers is a best-of-a-bad-job strategy relative to breeding, and helpers do not appreciably increase the reproductive output of the group in most years [20]. Adults may leave their natal territory to become breeders at another territory, but they do not retain helper status once they disperse [21]. Cobreeding groups in this population form when same-sex helpers disperse together to fill a breeding vacancy at an existing territory, or to establish a new territory. Helpers can also inherit a cobreeding position in their natal group by joining same-sex breeders (their same-sex parent along with any same-sex siblings) following the death or disappearance of all opposite-sex breeders coupled with the immigration of unrelated opposite-sex birds [22]. Thus, within social groups, breeders of both sexes are closely related to their same-sex cobreeders and typically are first-order kin [22,23]. Because it is uncommon for individuals to disperse after acquiring a breeding position, cobreeding groups usually persist until the death of one or more of the cobreeders.

Not all social groups contain cobreeders. Individuals can become single breeders by founding a new territory and attracting a mate, filling a reproductive vacancy as the sole breeder in an existing group, inheriting the natal territory as the sole breeder following the death or disappearance of all breeders within the group, or by attrition within a cobreeding coalition (disappearance of all but a single remaining breeder). Extra-group mating is rare in this population [24], so single breeders monopolize all reproduction within the group whereas cobreeders compete for parentage with other same-sex cobreeders in each nesting attempt.

Once established as a breeder, neither sex is challenged by larger same-sex coalitions—even as single breeders; rather, breeder turnovers generally occur only with the death of all breeders of one sex [25]. Breeder turnovers often involve 'power struggles', where competing coalitions of same-sex acorn woodpeckers (depending on the sex associated with the breeding vacancy) fight one another until one coalition succeeds. Power struggles may include dozens of birds from greater than 15 social groups and can last days or even weeks [26].

Territory quality for acorn woodpeckers at Hastings is determined primarily by the presence and size of granaries, specialized trees that are used to store acorns in individually drilled holes before the acorns are consumed. Territory quality, measured as granary size, remains relatively constant compared to annual fluctuations in acorn crops and breeding coalitions are often predicted primarily by the presence of granaries rather than the acorn crop in any particular year [17]. The exact number of functional holes in a group's granary, which may be cryptically spread out over dead limbs in the canopies of several trees, is difficult to count accurately. Furthermore, while

additional holes are added by group members on a more-or-less continuous basis, a major branch or entire granary may fall, episodically reducing granary size. Granary size is thus categorized as either low-quality (fewer than 2500 storage holes or high-quality (more than 2500 holes). All members of the social group participate in territory maintenance and defence [27].

## (b) Demographic methods and analyses

Active acorn woodpecker groups in the study area have been censused at approximately bimonthly intervals since 1972. Census data thus provide reliable estimates of when individuals joined or disappeared from a social group. We compiled data with group composition (i.e. number of breeders and helpers of each sex) and territory quality for each group for each year (as of 15 May, the peak of the breeding season) between 1974 and 2016.

## (c) Parentage assignment

Parentage was tested for fledglings hatched between 1984 and 2016. Blood was sampled from all birds for genotyping when captured, which was generally at nests or in roosting cavities as adults, or when banded as nestlings between 1984 and 2016 [28]. Blood was placed in Longmire's solution [29] and stored at −20°C onsite until subsequent DNA extraction and analysis. To assign parentage, we used 8–18 microsatellite loci developed for acorn woodpeckers from protocols modified from Armour *et al.* [30], Gibbs *et al.* [31] and Jones *et al.* [32]. Amplicons for each locus were produced in three multiplexed polymerase chain reactions (QIAGEN Multiplex Plus) and sized on an Applied Biosystems 3730 DNA analyser using Liz 500 as a molecular weight standard.

We tested the loci used in parentage assignments for deviations from Hardy–Weinberg equilibrium (HWE) and linkage disequilibrium using GenePop 4.7.5 [33] with 1000 dememorizations, 100 batches and 1000 iterations per batch. We ran the parentage assignment analyses for 5-year time periods (1990, 1995, 2000, 2005, 2010 and 2015) using 52–78 candidate parents in each year.

To reduce deviation in parentage assignment caused by presence of relatives, we first selected one male and one female candidate parent from each group (usually individuals with breeding status), and we then eliminated individuals that were first-order relatives based on parentage (i.e. usually this would be due to individuals that shared the same natal group). Of the 18 loci we commonly used in determining parentage, eight deviated from HWE in at least 1 year. Consequently, we examined assignments for all offspring, paying particular attention to loci that might have null alleles for a particular set of breeders (all individuals with breeding status in a social group for a given breeding attempt). Genotypes were either corrected by examining Mendelian transmission across generations for a particular allele, or deleted if we still suspected a null allele. We controlled for false discovery rate [34] in the linkage disequilibrium tests because of the large number of pairwise loci comparisons. No locus was in disequilibrium in more than 1 year; thus, we included all 18 loci in the parentage assignments.

Parentage was determined using CERVUS 3.0.7 [35] (simulation criteria: no. offspring = 10 000, proportion loci typed = 0.75, no. candidate mothers = 5, no. candidate fathers = 10, proportion candidate parents genotyped = 1.0, and proportion of loci mistyped = 0.02); we accepted assignments that produced at least 95% confidence for a single father–offspring dyad and excluded every possible male group member with breeding status within the previous 2 years. For all parental assignments, at most, two mismatches were allowed in the assigned parental–offspring triad. We did not necessarily exclude all possible helper–offspring dyads with 95% confidence (in the particular social group within the previous 2 years), but most (greater than 90%) could be excluded with 95% confidence

based on assigned parental triads. In some cases, a helper that was probably a full sibling of offspring could not be excluded because we had genotyped the helper at too few loci. Reproduction by helpers is extremely rare [36] and unlikely to alter the assignments used in the analysis presented here. For nests with cobreeders where not all offspring could be genotyped successfully, we conservatively partitioned the non-genotyped offspring equally among all cobreeders of a particular sex. Finally, to ensure that we did not bias our data to short-lived birds, we only examined parentage data for birds hatched up until 2006. For further details, see [22].

## (d) Breeder turnover index

To determine whether same-sex coalitions filled breeding vacancies more often than single breeders and, thus, whether cobreeding as a strategy may have helped individuals disperse successfully, we calculated a breeder turnover index for males by means of the methods used previously for females [22]. Briefly, for groups that experienced a turnover in breeders, we calculated the direction of change in the number of breeders; that is, whether the group had the same, more, or fewer breeders following the turnover. We calculated mean breeder turnover for low- and high-quality territories to test if larger cobreeder groups were more likely to win breeding vacancies at high-quality territories.

## (e) Statistical analyses

All analyses were conducted in R v. 4.0.2 [37]. General linear mixed models (LMM) and generalized linear mixed models (GLMM) were used for all analyses. In all models that included long-term individual fitness data, we used territory identity and individual identity as random factors, where appropriate. Package *lmerTest* 3.1-2 was used for mixed models [38]. Figures are plotted as box plots with raw data points in the dataset. Data points were jittered using the *geom_jitter* function in package *ggplot2* 3.3.3 to minimize the overlap of data points with the same values. Unless stated otherwise, values presented are means ± standard errors.

## (f) Hypotheses testing the direct fitness benefits of cobreeding

We investigated the effects of cobreeding on lifetime reproductive success (measured as lifetime young fledged) by grouping individuals by the mean number of cobreeders they bred with throughout their life (rounded up to the closest integer). For males, we lumped instances with 4 or more cobreeders because of the small number of large cobreeding coalitions. For every individual in the analysis, we quantified the following parameters (numbers in parentheses refer to models in electronic supplementary material, table S1): (i) young fledged per nesting attempt, the number of fledglings assigned to an individual in a given brood when the group fledged young successfully (see below); (ii) the age of first reproduction, the age in years when an individual of known age was first assigned parentage; (iii) reproductive lifespan, the number of years a bird was recorded as a putative breeder; (iv) the total number of nesting attempts over the individual's reproductive lifespan, where nesting attempts were defined as those reaching the incubation stage whether the bird in question gained parentage or not; and (v) lifetime reproductive success, the sum of all fledglings assigned to an individual over its lifetime. For each of these analyses, separate GLMMs were run for males and females, with number of breeders (1, 2, 3 or greater than or equal to 4 for males, and 1, 2 or 3 for females) and lifetime territory quality (low or high) as fixed predictors with territory identity as a random effect.

We next conducted analyses to test hypotheses about the demographic and ecological factors contributing to the

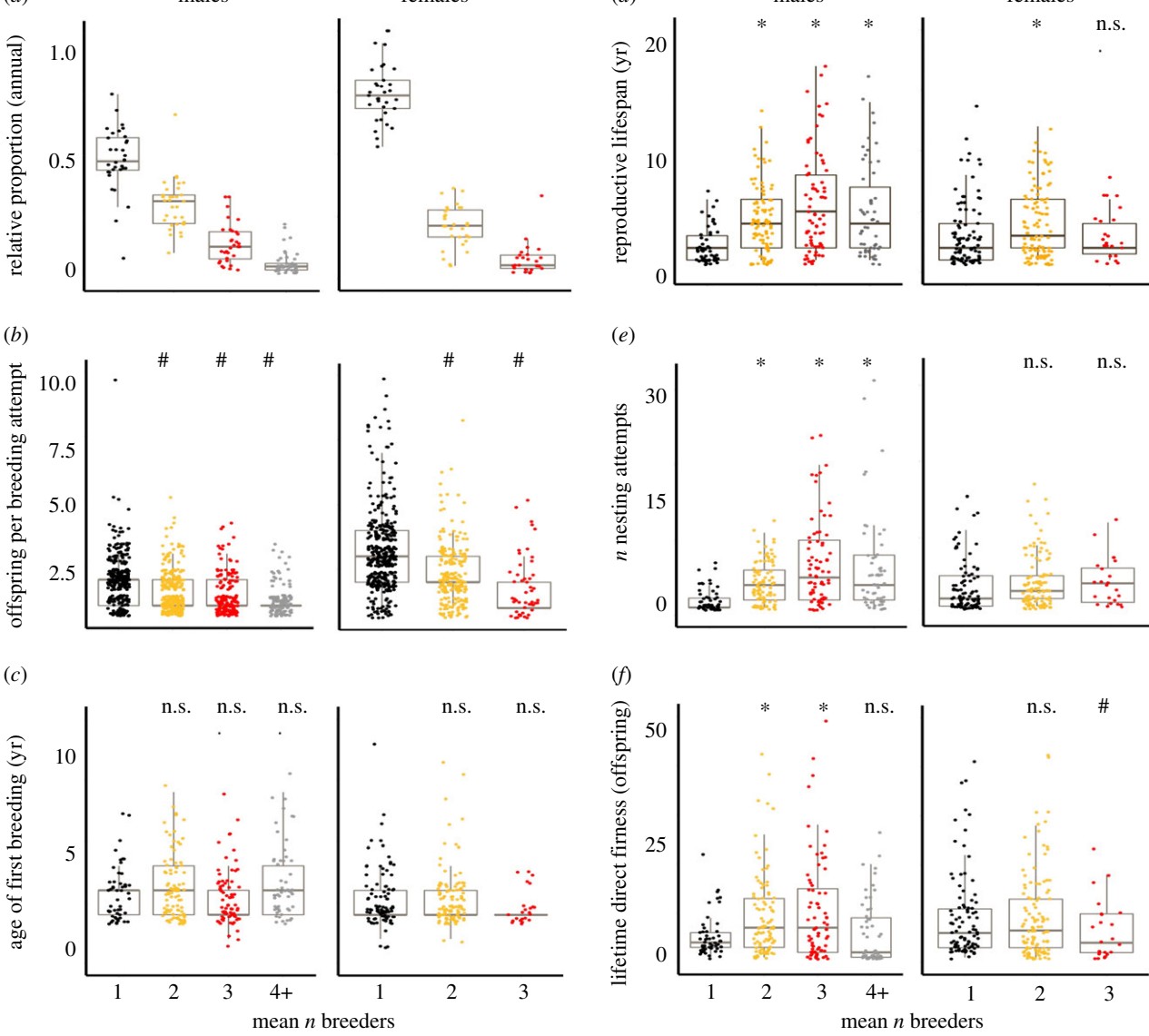

**Figure 1.** Individual fitness benefits as a function of the mean number of breeders (1 = single breeder) in male and female acorn woodpeckers (colours correspond to number of breeders). Boxplots denote 10th and 90th percentile of the data with the horizontal line showing the median. Raw jittered data points are shown. (*a*) The relative proportion of breeder composition per year by sex in the study population, across 43 years (1974–2016) of the continuous current study. In any given year, roughly half the males and approximately 70% of females were single breeders while the rest were in cobreeding coalitions; (*b–f*) show lifetime fitness data for birds from 1984 to 2006 (males *n* = 275, females *n* = 224). Figures show comparisons between single breeders versus cobreeding categories (mean number of cobreeders throughout an individual's life). Hash signs (#) show groups with fitness values significantly (*p* < 0.05) lower than single breeders. Asterisks (*) show groups with fitness values significantly higher than single breeders while 'n.s.' indicate groups with fitness values not significantly different than single breeders. (*b*) The number of fledglings produced by a breeder per successful nesting attempt for the group; (*c*) age of first breeding (when first assigned parentage); (*d*) reproductive lifespan (number of years as a breeder in the population); (*e*) number of nesting attempts that reached the incubation stage by the group during the tenure of a breeder; and (*f*) lifetime number of young (assigned to an individual via genetic parentage analysis) that reached fledgling stage. (Online version in colour.)

differences observed in the first set of analyses. We used GLMMs to ask whether (vi) the number of cobreeders differed on high-quality versus low-quality territories and (vii) a LMM to test whether the difference in coalition size before and after breeding group turnover differed on high-quality versus low-quality territories. Sample sizes and details for each model are given in the electronic supplementary material, table S1.

## 3. Results

### (a) Sample sizes and frequency of cobreeding

For males (*n* = 275), we obtained a mean of 4.4 ± 3.5 years of reproductive data and assigned parentage to 2264 fledglings in 1279 nesting attempts. For females (*n* = 224), we obtained

3.6 ± 2.2 years of reproductive data and assigned parentage to 2004 fledglings in 837 nesting attempts. For males, the lifetime mean coalition size ranged from 1 to 8 (single breeders: *n* = 52; cobreeding duos: *n* = 95; cobreeding trios: *n* = 73; four or more cobreeders: *n* = 55). For females, the lifetime coalition size ranged from 1 to 3 (single breeders: *n* = 91; duos: *n* = 110; and trios: *n* = 23). Overall, cobreeding was more common for males than for females (in any given year, 55.8% of males shared reproduction with at least one cobreeder, whereas only 31.3% of females shared reproduction; figure 1*a*).

### (b) Lifetime reproductive output

After controlling for territory quality, cobreeders of both sexes produced fewer offspring per nesting attempt than did

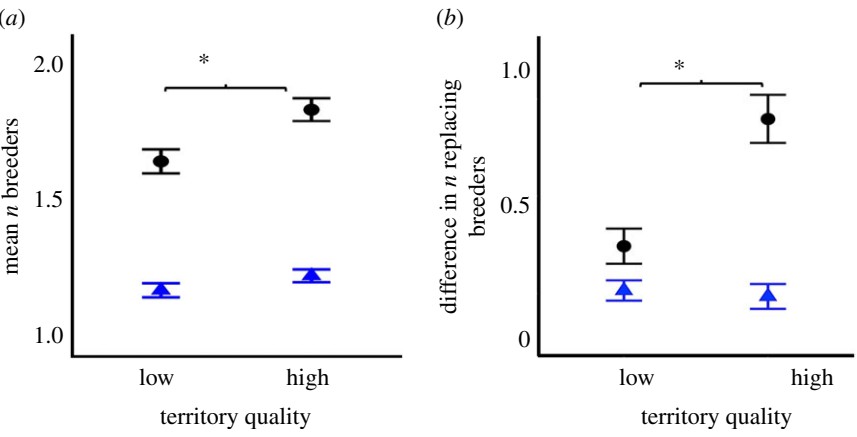

**Figure 2.** Association between cobreeding and territory quality, in acorn woodpeckers. Circles denote males, triangles denote females. Error bars indicate standard errors, and the asterisks represent statistically significant differences ($p < 0.05$) between low- and high-quality territories. (*a*) Mean number of male (circles) and female (triangles) breeders at low- and high-quality territories (males $n = 1442$ group years, females $n = 1405$ group years). (*b*) Mean breeder turnover index (positive values = replaced by a larger number of males/females during complete breeder turnover, 0 = replaced by the same number of males/females) as a function of territory quality (males $n = 282$ turnovers, females $n = 453$ turnovers). (Online version in colour.)

single breeders (figure 1*b*; electronic supplementary material, table S1). The age at first reproduction was not influenced by the number of cobreeders in either sex (figure 1*c*; electronic supplementary material, table S2). For males, reproductive lifespan (years) was significantly higher ($p < 0.05$) for coalitions of all sizes compared to singletons (figure 1*d*; electronic supplementary material, table S3). Cobreeder males in all coalition sizes also had more lifetime breeding attempts than single breeding males (figure 1*e*; electronic supplementary material, table S4). Males in duos and trios had higher lifetime reproductive success than single breeders (figure 1*f*; electronic supplementary material, tables S1 and S5). For females, cobreeding also affected reproductive lifespan (figure 1*d*; electronic supplementary material, table S3). Female duos had longer reproductive lifespans than single breeders but cobreeding did not affect the number of lifetime reproductive attempts (figure 1*e*; electronic supplementary material, table S4). As a result, single breeding females had significantly higher lifetime reproductive success than females in trios, but not compared to duos (figure 1*f*; electronic supplementary material, table S5). For males, high-quality territories significantly increased (i) the number of offspring per successful nest attempt, (ii) reproductive lifespan, (iii) lifetime breeding attempts and (iv) lifetime reproductive success. In females, high-quality territories did not significantly increase any fitness response variables. For the results of a model investigating the interaction between number of cobreeders and territory quality, see the electronic supplementary material, table S1.

### (c) Territory quality and inter-group conflict

Cobreeding males were more likely to occur on high-quality territories than were single breeders (figure 2*a*; electronic supplementary material, table S6). For females, however, there was no association between the number of cobreeders and territory quality (electronic supplementary material, table S6). We found that reproductive vacancies of breeder males, but not females, were filled by significantly larger coalitions on high-quality territories relative to low-quality territories (figure 2*b*; electronic supplementary material, table S7).

## 4. Discussion

The benefits of cobreeding differed between the sexes in the same population. Male acorn woodpeckers in cobreeding coalitions experienced longer reproductive lifespans and engaged in more nesting attempts than single male breeders, leading to male duos and trios having significantly greater lifetime reproductive success (relative to groups with a single breeder). Females, in contrast, did not accrue similar benefits from cobreeding. Female cobreeding duos had significantly longer reproductive lifespans and comparable lifetime reproductive success to single breeders but trios had significantly lower lifetime reproductive success relative to single breeders. For cobreeding male duos and trios, reproductive advantages such as an increased reproductive lifespan and number of nesting attempts over their lifetime compensated for any direct fitness losses within nesting attempts. In females, longer reproductive lifespans led to lifetime reproductive success in duos that was equivalent to single breeders. Thus, for cobreeding male duos and trios, and female duos (the most common coalition sizes for those respective sexes outside of singletons), lifetime reproductive success alone may provide sufficient fitness benefits to explain the presence of cooperative polygamy, since lifetime reproductive success is greater than or equivalent to single breeders.

Previous research on this population suggests that these sex-specific differences in lifetime reproductive success are likely driven by at least two factors: (i) sex-specific physiological costs of cobreeding and (ii) sex-specific interactions between territory quality and the likelihood of cobreeding. First, the energetic costs of cobreeding appear to be substantially higher for females than for males. Cobreeding female acorn woodpeckers compete for reproduction by destroying each other's eggs: each female typically removes eggs laid in the communal nest until she has laid her own first egg [39]. This behaviour results in synchrony of egg-laying and an equal partitioning of reproduction among cobreeding females [40]. However, egg destruction by cobreeding females reduces the eggs contributed by each female and is responsible for the loss of up to 38% of all eggs laid in communal nests; the associated energetic costs are thought to be an important constraint on the occurrence of female cobreeding [40]. Despite these constraints, cobreeding female duos seem to offset costs

associated with any particular nesting attempt by having longer reproductive lifespans and thus lifetime number of young produced that equals that of single breeders. By contrast, cobreeding males compete primarily by mate guarding [41,42], which is presumably less energetically costly than egg destruction and thus may explain the apparently larger benefits of cobreeding in males.

Several lines of evidence suggest that propensity to form cobreeding coalitions is related to increased territory quality for males, but not for females. Because drilling granary holes is time-consuming (approx. 30 min per hole, or approx. 500 h of drilling for a relatively small granary), a group cannot create a large granary in a single season, or even a year; rather, granaries accumulate over many years and across generations of woodpeckers [18]. Territory quality (i.e. granary size) is an important predictor of reproductive success [20]. For females, cobreeding is not more likely to occur on high-quality territories [22]. By contrast, we found that groups on high-quality territories were more likely to contain cobreeding rather than single breeder males.

This relationship between the propensity to cobreed and territory quality may arise for two reasons. First, high-quality territories with larger granaries support more non-breeding helpers for longer periods of time [43]. Male acorn woodpeckers are philopatric and tend to stay back as helpers more than females [44]. Consequently, males are more likely than females to act as helpers and ultimately to inherit a breeding position and share reproduction with their same-sex parent and all male siblings who may also be present as helpers [45]. Second, coalitions of males can fill high-quality territories by winning inter-group contests against neighbouring groups. Large coalitions tend to outcompete small coalitions during power struggles [25]. Consistent with this pattern, we found that reproductive vacancies of breeder males, but not females, were filled by significantly larger coalitions on high-quality territories relative to low-quality territories. However, it is also important to recognize that the interactions between territory quality, group size and resource availability each year are complex, and territory quality may affect both male and female fitness in more subtle ways that we could not resolve with this dataset.

Why, then, do females sometimes share reproduction (especially trios), and why do males ever breed alone or with four or more males? Previous research on this population determined that cobreeding is a best-of-a-bad-job strategy for females, driven by competition for reproductive opportunities [22]. However, we found that cobreeding female duos had direct lifetime reproductive success comparable to single breeders. Given that female cobreeders are typically closely related, the total inclusive fitness benefits of cobreeding in duos are probably greater than previously assumed and thus both direct and indirect fitness may influence cobreeding by females.

An individual's decision to breed with or without cobreeders is likely a function of several factors that include natal territory quality [46], the number of same-sex siblings in the natal group that represent potential cobreeders (or that could form a coalition if a reproductive vacancy were to occur elsewhere), the degree of habitat saturation in the population, and the distance and quality of territories with reproductive vacancies [47]. For example, individuals may choose to remain and cobreed at their natal high-quality territory when they inherit the territory, versus leaving the

group and attempting to disperse alone. Males may disperse to low-quality territories alone but likely need to be a part of a coalition to compete for reproductive vacancies on high-quality territories. Single breeding males tend to be on low-quality territories where they can monopolize each reproductive attempt, but low-quality territories may not be viable breeding sites in poor acorn crop years. On the other hand, cobreeding in males is more common at high-quality territories where each male may have fewer offspring per reproductive attempt; but have longer reproductive tenures, more lifetime reproductive attempts, and higher direct lifetime reproductive success. For females, cobreeding duos also benefit from cobreeding, with an increase in reproductive lifespan and an equivalent lifetime reproductive output in duos compared to single breeding females despite the lack of an advantage attributed to higher territory quality. However, the higher physiological costs of cobreeding for females (e.g. egg destruction) may negate any potential benefits among cobreeding trios.

## 5. Conclusion

Cooperative polygamy has often been considered a lifetime reproductive compromise. Here we show that, for male acorn woodpeckers, the lifetime reproductive success of birds cobreeding as a duo or trio may be sufficient to compensate for the annual reduction in reproductive output they suffer by sharing reproduction. Thus, the reproductive fitness benefit by itself may be sufficient to explain the occurrence of this strategy in male acorn woodpeckers without invoking inclusive fitness benefits [6]. Adult acorn woodpeckers of both sexes typically cobreed with first-order kin; the total fitness benefits of cobreeding (direct + inclusive) may be even greater. An analysis incorporating inclusive fitness benefits of helping and cobreeding is beyond the scope of this study but, when completed, will further our understanding of why cooperative strategies are prevalent in this population.

These results underscore the importance of lifetime, longitudinal data on natural populations to better understand the evolution of seemingly disadvantageous cooperative behaviours. Longitudinal data can also reveal that similar reproductive strategies in males and females may be shaped by very different selective pressures.

Ethics. All relevant research and research ethics certification was acquired before the research was initiated.

Data accessibility. We have made the data available as electronic supplementary material.

Authors' contributions. S.B.: conceptualization, data curation, formal analysis, investigation, methodology, visualization, writing-original draft, writing-review and editing; C.R.: conceptualization, formal analysis, methodology, writing-original draft and writing-review and editing; E.L.W.: conceptualization, data curation, funding acquisition, project administration, resources, supervision, writing-review and editing; J.H.: data curation, formal analysis, funding acquisition, methodology, project administration, resources, software, writing-review and editing; H.L.D.: data curation, formal analysis, funding acquisition, investigation, project administration, resources, software, writing-review and editing; W.D.K.: conceptualization, data curation, formal analysis, funding acquisition, investigation, methodology, project administration, resources, software, supervision, validation, writing-review and editing.

All authors gave final approval for publication and agreed to be held accountable for the work performed therein.

Competing interests. We declare we have no competing interests.

**Funding.** This study was funded by the National Science Foundation Division of Environmental Biology (grant no. 1256394) and Division of Integrative Organismal Systems (grant nos. 1455881 and 1455900). H.L.D. was supported by a Royal Society Research Grant (RG170425) and a NERC Fellowship (NE/I021748/1).

**Acknowledgements.** We thank Hastings Natural History Reservation for decades of help with logistics, as well as the 162 field assistants, post-docs, and graduate students who have helped on this project. We also thank M. Festa-Bianchet, an anonymous reviewer and an anonymous associate editor for constructive feedback on this manuscript.

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
