## [Peer Review File · Proceedings of the Royal Society B: Biological Sciences]

Review History

RSPB-2021-0579.R0 (Original submission)

Review form: Reviewer 1 (Marco Festa-Bianchet)

Recommendation

Major revision is needed (please make suggestions in comments)

Scientific importance: Is the manuscript an original and important contribution to its field?

Excellent

General interest: Is the paper of sufficient general interest?

Excellent

Quality of the paper: Is the overall quality of the paper suitable?

Good

Is the length of the paper justified?

Yes

Should the paper be seen by a specialist statistical reviewer?

No

Do you have any concerns about statistical analyses in this paper? If so, please specify them explicitly in your report.

No

It is a condition of publication that authors make their supporting data, code and materials available - either as supplementary material or hosted in an external repository. Please rate, if applicable, the supporting data on the following criteria.

Is it accessible?

Yes

Is it clear?

Yes

Is it adequate?

Yes

Do you have any ethical concerns with this paper?

No

Comments to the Author

I enjoyed reading this paper. It is very well written, examines several interesting questions, the long-term database is remarkable and the sample size formidable. The results are well presented and will be of interest to a broad readership. They clearly show an advantage to male duos and no advantage to female duos. Just as clearly, male coalitions are associated with better territories and better territories lead to greater lifetime reproductive success. So, do coalitions themselves provide benefits independently of territory quality? That question is not addressed here, yet an analysis of lifetime success that accounts for territory quality could easily be added to this paper. If there is no advantage to duos once territory size is accounted for, then the only apparent advantage of coalitions would be access to better territories.

Specific comments:

L. 123-15: Sorry, I'm confused. The helper joins a parent of the same sex when all opposite-sex breeders have disappeared: does that not lead to a unisex group?

L. 160-163: Genotyping from 1974? Microsatellites were discovered about 1989. Were blood samples collected and stored?

L. 173-176: I'm confused here as well, can this be explained better?

L. 205-207: At this point it is not totally clear what happens when only one member of a co-breeder pair disappears. Is that considered a turnover?

L. 236: is 'nestling' here the same as 'fledgling' a few lines above?

L. 271-273: 8.67 is more than 8.58, the results of some of these tests comparing 3-4 groups need to be better explained.

Most of P. 13: These results strongly suggest that there should be an analysis of the combined effects of number of cobreeders and territory quality.

L. 285 : Extra “was”.

L. 367-379: Not sure I understand this - territory quality was determined by granary size, so what is the effect of a poor crop of acorns? More holes will not compensate for fewer acorns?

Fig. 1 panel A: The Y-axis label (frequency of cooperation) appears incorrect. Panel B: If there is a slash between offspring and breeding, I cannot see it.

Marco Festa-Bianchet

Review form: Reviewer 2

Recommendation

Accept with minor revision (please list in comments)

Scientific importance: Is the manuscript an original and important contribution to its field?

Excellent

General interest: Is the paper of sufficient general interest?

Excellent

Quality of the paper: Is the overall quality of the paper suitable?

Good

Is the length of the paper justified?

Yes

Should the paper be seen by a specialist statistical reviewer?

No

Do you have any concerns about statistical analyses in this paper? If so, please specify them explicitly in your report.

Yes

It is a condition of publication that authors make their supporting data, code and materials available - either as supplementary material or hosted in an external repository. Please rate, if applicable, the supporting data on the following criteria.

Is it accessible?

Yes

Is it clear?

Yes

Is it adequate?

No

Do you have any ethical concerns with this paper?

No

Comments to the Author

This is a very interesting, well-written paper, which is remarkably error-free. It tackled the fitness costs and benefits of a rare social system where separate coalitions of males and females combine

to breed in a polygynandrous association, and cooperate to rear a joint clutch. The data are based on an exemplary long-term study which is one of the best ever accumulated for a free-living population. The data are particularly remarkable because of features not mentioned in the paper, such as the difficulty in accessing nests high in trees.

The results are of great interest. While ‘cooperation’ appears to have several attendant costs, in the modal coalition for males (two) birds have higher lifetime reproductive success than birds that breed solitarily because they have longer reproductive spans and more breeding attempts. However, for females there are no benefits from being in a coalition. The authors hypothesise several reasons why this might be true

There are a few things that might have attracted discussion but are not addressed as much as I would like at the moment. First, while duos of males do best, the outcomes for larger coalitions are very poor (which are not infrequent). The authors present evidence that larger male coalitions may be better at taking over (and presumably defending) territories and its vital granary resource, which might explain a suboptimal group size, but I think this point requires further attention. Second, and following on from this, having a high quality territory dramatically increases success (Table S5). However, the relevant models treat coalition size and territory quality as variables that do not interact. Although I generally admire parsimony in statistical analysis, in this case an interaction between coalition size and quality seems both biologically possible (eg Fig 2b) and potentially informative. Third, the explanation that large group size facilitates territory acquisitions seems remarkably similar to arguments that have already been developed for cooperatively polyandrous societies such as the Galapagos hawk where only one female breeds, and this might be mentioned. Last, I have always thought that cooperative polygynandry and polyandry must be quite different, but in this case the difference seems to stem from the ability of both males and females in polygynandrous societies to achieve the optimum.

Fig. 1 - Is it really true as suggested in the caption that the horizontal lines within the boxes represent the mean? It is conventional for the median to be depicted, and this is suggested by what appear to be identical values in some of the plots (eg C and D). Similarly, the whiskers often depict the 10 and 90th percentiles rather than quartiles. The points beyond the whiskers also imply this is also true.

Decision letter (RSPB-2021-0579.R0)

06-Apr-2021

Dear Dr Barve:

Your manuscript has now been peer reviewed and the reviews have been assessed by an Associate Editor. The reviewers’ comments (not including confidential comments to the Editor) and the comments from the Associate Editor are included at the end of this email for your reference. As you will see, the reviewers and the Editors have raised some concerns with your manuscript and we would like to invite you to revise your manuscript to address them.

Research ethics:

Use of animals and field studies:

It is a condition of publication that you make available the data and research materials supporting the results in the article. Please see our Data Sharing Policies (<https://royalsociety.org/journals/authors/author-guidelines/#data>). Datasets should be deposited in an appropriate publicly available repository and details of the associated accession number, link or DOI to the datasets must be included in the Data Accessibility section of the article (<https://royalsociety.org/journals/ethics-policies/data-sharing-mining/>). Reference(s) to datasets should also be included in the reference list of the article with DOIs (where available).

All supplementary materials accompanying an accepted article will be treated as in their final form. They will be published alongside the paper on the journal website and posted on the online

figshare repository. Files on figshare will be made available approximately one week before the accompanying article so that the supplementary material can be attributed a unique DOI. Please try to submit all supplementary material as a single file.

Please submit a copy of your revised paper within three weeks. If we do not hear from you within this time your manuscript will be rejected. If you are unable to meet this deadline please let us know as soon as possible, as we may be able to grant a short extension.

Best wishes,
Dr Maurine Neiman
mailto:proceedingsb@royalsociety.org

Associate Editor
Board Member: 1
Comments to Author:

This manuscript was seen by two reviewers, who are broadly congruent in their assessment, they found very much to like, and I concur. The manuscript uses a large dataset to test topical and broad hypotheses, and it is well written. Both reviewers highlighted one important issue, namely the potential effects of territory quality, that need to be better incorporated into the analyses, in interaction with the number of co-breeders/coalition size. Both reviewers have further minor comments, notably Figure 1 needs a revision and reviewer 2 gives some helpful suggestions on referring to previous literature and interpretation of the results. None of these revision seem particularly arduous, although the outcome of the additional analyses may of course generate some further results and refinement of the discussion. I look forward to the revised version.

Reviewer(s)' Comments to Author:

Referee: 1

Comments to the Author(s)

I enjoyed reading this paper. It is very well written, examines several interesting questions, the long-term database is remarkable and the sample size formidable. The results are well presented and will be of interest to a broad readership. They clearly show an advantage to male duos and no advantage to female duos. Just as clearly, male coalitions are associated with better territories and better territories lead to greater lifetime reproductive success. So, do coalitions themselves provide benefits independently of territory quality? That question is not addressed here, yet an analysis of lifetime success that accounts for territory quality could easily be added to this paper. If there is no advantage to duos once territory size is accounted for, then the only apparent advantage of coalitions would be access to better territories.

Specific comments:

L. 123-15: Sorry, I'm confused. The helper joins a parent of the same sex when all opposite-sex breeders have disappeared: does that not lead to a unisex group?

L. 160-163: Genotyping from 1974? Microsatellites were discovered about 1989. Were blood samples collected and stored?

L. 173-176: I'm confused here as well, can this be explained better?

L. 205-207: At this point it is not totally clear what happens when only one member of a co-breeder pair disappears. Is that considered a turnover?

L. 236: is 'nestling' here the same as 'fledgling' a few lines above?

L. 271-273: 8.67 is more than 8.58, the results of some of these tests comparing 3-4 groups need to be better explained.

Most of P. 13: These results strongly suggest that there should be an analysis of the combined effects of number of cobreeders and territory quality.

L. 285 : Extra "was".

L. 367-379: Not sure I understand this - territory quality was determined by granary size, so what is the effect of a poor crop of acorns? More holes will not compensate for fewer acorns?

Fig. 1 panel A: The Y-axis label (frequency of cooperation) appears incorrect. Panel B: If there is a slash between offspring and breeding, I cannot see it.

Marco Festa-Bianchet

Referee: 2

Comments to the Author(s)

This is a very interesting, well-written paper, which is remarkably error-free. It tackled the fitness costs and benefits of a rare social system where separate coalitions of males and females combine to breed in a polygynandrous association, and cooperate to rear a joint clutch. The data are based on an exemplary long-term study which is one of the best ever accumulated for a free-living population. The data are particularly remarkable because of features not mentioned in the paper, such as the difficulty in accessing nests high in trees.

The results are of great interest. While 'cooperation' appears to have several attendant costs, in the modal coalition for males (two) birds have higher lifetime reproductive success than birds that breed solitarily because they have longer reproductive spans and more breeding attempts. However, for females there are no benefits from being in a coalition. The authors hypothesise several reasons why this might be true

There are a few things that might have attracted discussion but are not addressed as much as I would like at the moment. First, while duos of males do best, the outcomes for larger coalitions are very poor (which are not infrequent). The authors present evidence that larger male coalitions may be better at taking over (and presumably defending) territories and its vital granary resource, which might explain a suboptimal group size, but I think this point requires further attention. Second, and following on from this, having a high quality territory dramatically increases success (Table S5). However, the relevant models treat coalition size and territory quality as variables that do not interact. Although I generally admire parsimony in statistical analysis, in this case an interaction between coalition size and quality seems both biologically possible (eg Fig 2b) and potentially informative. Third, the explanation that large group size facilitates territory acquisitions seems remarkably similar to arguments that have already been developed for cooperatively polyandrous societies such as the Galapagos hawk where only one female breeds, and this might be mentioned. Last, I have always thought that cooperative polygynandry and polyandry must be quite different, but in this case the difference seems to stem from the ability of both males and females in polygynandrous societies to achieve the optimum.

Fig. 1 - Is it really true as suggested in the caption that the horizontal lines within the boxes represent the mean? It is conventional for the median to be depicted, and this is suggested by what appear to be identical values in some of the plots (eg C and D). Similarly, the whiskers often depict the 10 and 90th percentiles rather than quartiles. The points beyond the whiskers also imply this is also true.

Author's Response to Decision Letter for (RSPB-2021-0579.R0)

See Appendix A.

RSPB-2021-0579.R1 (Revision)

Review form: Reviewer 1 (Marco Festa-Bianchet)

Recommendation

Accept with minor revision (please list in comments)

Scientific importance: Is the manuscript an original and important contribution to its field?

Excellent

General interest: Is the paper of sufficient general interest?

Excellent

Quality of the paper: Is the overall quality of the paper suitable?

Excellent

Is the length of the paper justified?

Yes

Should the paper be seen by a specialist statistical reviewer?

No

Do you have any concerns about statistical analyses in this paper? If so, please specify them explicitly in your report.

No

It is a condition of publication that authors make their supporting data, code and materials available - either as supplementary material or hosted in an external repository. Please rate, if applicable, the supporting data on the following criteria.

Is it accessible?

Yes

Is it clear?

Yes

Is it adequate?

Yes

Do you have any ethical concerns with this paper?

No

Comments to the Author

I liked this paper when I first read it and I like it even better now. Excellent analysis, impressive sample size, reasonable interpretations. I have a few final suggestions:

L. 35: of nesting attempts (?).

L. 279: Either remove the 'either' or add 'or males in groups of 4 and larger'.

Fig. 1: I do not understand A, partly because the axis label and the legend description do not match. Does the figure mean that about half of all males and 3/4 of females were single breeders?

Fig. 2: I am not totally sure I understand the natural history and some definitions here - presumably the 'initial' coalition size is the maximum during its history? If I understood correctly, established breeders are not challenged, so breeder turnover presumably is nearly always when the last bird dies, so that at that point there is one breeder left, even if earlier on there may have been 2 or more co-breeders?

Marco Festa-Bianchet

Decision letter (RSPB-2021-0579.R1)

23-Jun-2021

Dear Dr Barve:

Your manuscript has now been peer reviewed and the reviews have been assessed by an Associate Editor. The reviewer's comments (not including confidential comments to the Editor) and the comments from the Associate Editor are included at the end of this email for your reference. As you will see, the reviewer and the Editor have raised some concerns with your manuscript and we would like to invite you to revise your manuscript to address them. In particular, the Associate Editor provided a very thorough and well justified explanation for why she wants to see some additional revisions above and beyond those requested by the reviewer.

When submitting your revision please upload a file under "Response to Referees" in the "File Upload" section. This should document, point by point, how you have responded to the reviewers' and Editors' comments, and the adjustments you have made to the manuscript. We

require a copy of the manuscript with revisions made since the previous version marked as 'tracked changes' to be included in the 'response to referees' document.

Research ethics:

Use of animals and field studies:

It is a condition of publication that you make available the data and research materials supporting the results in the article (<https://royalsociety.org/journals/authors/author-guidelines/#data>). Datasets should be deposited in an appropriate publicly available repository and details of the associated accession number, link or DOI to the datasets must be included in the Data Accessibility section of the article (<https://royalsociety.org/journals/ethics-policies/data-sharing-mining/>). Reference(s) to datasets should also be included in the reference list of the article with DOIs (where available).

Online supplementary material will also carry the title and description provided during submission, so please ensure these are accurate and informative. Note that the Royal Society will not edit or typeset supplementary material and it will be hosted as provided. Please ensure that

the supplementary material includes the paper details (authors, title, journal name, article DOI). Your article DOI will be 10.1098/rspb.[paper ID in form xxxx.xxxx e.g. 10.1098/rspb.2016.0049].

Please submit a copy of your revised paper within three weeks. If we do not hear from you within this time your manuscript will be rejected. If you are unable to meet this deadline please let us know as soon as possible, as we may be able to grant a short extension.

Best wishes,
Dr Maurine Neiman
Editor, Proceedings B
mailto: proceedingsb@royalsociety.org

Associate Editor
Board Member: 1
Comments to Author:

The authors have revised this manuscript, including additional analyses. One of the original reviewers looked through this revision, and had a few suggestions and queries remaining. Upon going through the revisions, I noted a few -mostly minor- additional issues that should be addressed in a revised manuscript and revised response to reviewer comments.

First, it is not well argued why territory quality is quantified as two levels, which is easiest for the analysis of interactions. But these interactions are not included and results are described as if territory quality is continuous, e.g. in the supplemental tables the effect size of territory quality is given without reference category (i.e. implying it is a continuous variable) and at Question 6, "increasing territory quality"). This needs to be clarified, and can be solved - for example - by a short rationale at l. 148-149 why territory quality has two levels rather than continuous variation and slight tweaking of the results descriptions and clarifying in the supplemental tables output and legend how territory quality is defined.

Second, sample sizes have changed in several models, and outcomes have changed qualitatively and quantitatively (e.g. Supplementary Table questions 3, 4 and 5; table S3, reproductive lifespan, large differences for males and duo females). Why this occurred is however not addressed in the response to reviewers, so this should be briefly addressed in a revised response. Because of these additional changes, not all changes are tracked in the revised manuscript (response to reviewers document) and supplemental files. To gain a complete overview of changes, I request the authors re-submit a full comparison outlining all differences between the original manuscript (at first submission) and the final version (incorporating all previously made changes as well as all changes and suggestions requested here).

Third, regarding the discussion of the new result that female duos have similar lifetime reproductive success to single females (in the previous manuscript version their lifetime reproductive success was lower): I am not sure I follow, if single females and duos have similar lifetime reproductive success, why this alone can explain cooperative polygamy in females (see l. 310-311; similar statement is made in Abstract), why it thus 'reveals a potential driver of cobreeding in females' (l. 363), and why you conclude 'cobreeding duos seem to benefit in similar ways to males' (l. 380-381). It might be good to expand this argument a little.

Fourth, in addition to the query from the reviewer about Figure 1: the asterisks are confusing, it seem they can denote two different things? For example, Figure 1B the asterisks do not seem to concur with the results of model S1, since 2, 3 and 4 co-breeders are all different from N=1 (similar to Table S3, but panel Figure 1D shows 3 asterisks, not 1; likewise, in panel 1F the asterisk does not seem to concur with the output of model S5 -

Related to this, in l. 524-525: error in text as to what asterisks denote and text is confusing "...and asterisks denote model estimates where which are significantly greater than for single breeders. Asterisk over a 'mean N breeder' category indicates it has model estimate that differed significantly from the reference category of single breeder".

L. 313-314, word missing or redundancy here? “sex-specific differences in lifetime reproductive success between males and females”

Reviewer(s)' Comments to Author:

Referee: 1

Comments to the Author(s)

I liked this paper when I first read it and I like it even better now. Excellent analysis, impressive sample size, reasonable interpretations. I have a few final suggestions:

L. 35: of nesting attempts (?).

L. 279: Either remove the 'either' or add 'or males in groups of 4 and larger'.

Fig. 1: I do not understand A, partly because the axis label and the legend description do not match. Does the figure mean that about half of all males and 3/4 of females were single breeders?

Fig. 2: I am not totally sure I understand the natural history and some definitions here - presumably the 'initial' coalition size is the maximum during its history? If I understood correctly, established breeders are not challenged, so breeder turnover presumably is nearly always when the last bird dies, so that at that point there is one breeder left, even if earlier on there may have been 2 or more co-breeders?

Marco Festa-Bianchet

Author's Response to Decision Letter for (RSPB-2021-0579.R1)

See Appendix B.

Decision letter (RSPB-2021-0579.R2)

19-Jul-2021

Dear Dr Barve

I am pleased to inform you that your manuscript RSPB-2021-0579.R2 entitled "Lifetime reproductive benefits of cooperative polygamy vary for males and females in the acorn woodpecker (*Melanerpes formicivorus*)" has been accepted for publication in Proceedings B.

I do suggest some minor revisions to your manuscript. In particular, in the interest of transparency, we request that you add the full details of the models with the territory quality interactions to the supplement and refer to these analysis outcomes in the results. Because the schedule for publication is very tight, it is a condition of publication that you submit the revised version of your manuscript within 7 days. If you do not think you will be able to meet this date please let us know.

To revise your manuscript, log into <https://mc.manuscriptcentral.com/prsb> and enter your Author Centre, where you will find your manuscript title listed under "Manuscripts with Decisions." Under "Actions," click on "Create a Revision." Your manuscript number has been appended to denote a revision. You will be unable to make your revisions on the originally

submitted version of the manuscript. Instead, revise your manuscript and upload a new version through your Author Centre.

If you wish to submit your data to Dryad (<http://datadryad.org/>) and have not already done so you can submit your data via this link [http://datadryad.org/submit?journalID=RSPB&manu=\(Document not available\)](http://datadryad.org/submit?journalID=RSPB&manu=(Document not available)) which

will take you to your unique entry in the Dryad repository. If you have already submitted your data to dryad you can make any necessary revisions to your dataset by following the above link. Please see <https://royalsociety.org/journals/ethics-policies/data-sharing-mining/> for more details.

Sincerely,
 Dr Maurine Neiman
 Editor, Proceedings B
<mailto:proceedingsb@royalsociety.org>

Associate Editor:

Comments to Author:

The authors presented a clear response to all queries raised, and amended the manuscript. I agree that the revisions have improved the clarity of the manuscript.

Author's Response to Decision Letter for (RSPB-2021-0579.R2)

See Appendix C.

Decision letter (RSPB-2021-0579.R3)

22-Jul-2021

Dear Dr Barve

I am pleased to inform you that your manuscript entitled "Lifetime reproductive benefits of cooperative polygamy vary for males and females in the acorn woodpecker (*Melanerpes formicivorus*)" has been accepted for publication in Proceedings B.

Your article has been estimated as being 8 pages long. Our Production Office will be able to confirm the exact length at proof stage.

Data Accessibility section

Open Access

Paper charges

Sincerely,

Proceedings B

Appendix A

Dear Editor and Associate Editor,

Thank you for the constructive feedback we received on our Manuscript ID RSPB-2021-0579 “Lifetime direct fitness consequences of cooperative polygamy vary for males and females in the acorn woodpecker (*Melanerpes formicivorus*)”. The revisions suggested by the reviewers and editor were encouraging and improved the quality of the manuscript. We have addressed the reviewers’ comments in the main text and modified the analyses based on the reviewer and associate editor’s comments. A few results have changed but the overall findings remain unchanged. We thank you again and hope that these revisions are satisfactory. The responses to comments are listed below and any changes made to the main text (see below for entire manuscript) are in blue.

Sincerely,

Sahas Barve

Associate Editor

Board Member: 1

Comments to Author:

This manuscript was seen by two reviewers, who are broadly congruent in their assessment, they found very much to like, and I concur. The manuscript uses a large dataset to test topical and broad hypotheses, and it is well written. Both reviewers highlighted one important issue, namely the potential effects of territory quality, that need to be better incorporated into the analyses, in interaction with the number of co-breeders/coalition size. Both reviewers have further minor comments, notably Figure 1 needs a revision and reviewer 2 gives some helpful suggestions on referring to previous literature and interpretation of the results. None of these revision seem particularly arduous, although the outcome of the additional analyses may of course generate some further results and refinement of the discussion. I look forward to the revised version.

Reviewer(s)' Comments to Author:

Thank you for your positive comments on the manuscript. We have revised analyses based on your suggestions and made other changes you have suggested throughout the manuscript.

Referee: 1

Comments to the Author(s)

I enjoyed reading this paper. It is very well written, examines several interesting questions, the long-term database is remarkable and the sample size formidable. The results are well presented and will be of interest to a broad readership. They clearly show an advantage to male duos and no advantage to female duos. Just as clearly, male coalitions are associated with better territories and better territories lead to greater lifetime reproductive success. So, do coalitions themselves provide benefits

independently of territory quality? That question is not addressed here, yet an analysis of lifetime success that accounts for territory quality could easily be added to this paper. If there is no advantage to duos once territory size is accounted for, then the only apparent advantage of coalitions would be access to better territories.

Thank you for your positive comments on the manuscript. We have revised analyses based on your suggestions and made other changes you have suggested.

Specific comments:

L. 123-15: Sorry, I'm confused. The helper joins a parent of the same sex when all opposite-sex breeders have disappeared: does that not lead to a unisex group?

Helpers become cobreeders with their same-sex parents when their opposite-sex parents disappear (and unrelated opposite-sex birds move in to become the new breeders). It is hence not a unisex group. We have modified that statement in the manuscript to remove the ambiguity. Please see lines 125-126 in the revised manuscript.

Revised statement- *Helpers can also inherit a breeding position in their natal group by joining same-sex breeders (their parent and his or her cobreeders) following the death or disappearance of all opposite-sex breeders and immigration of unrelated opposite-sex birds [22]*

L. 160-163: Genotyping from 1974? Microsatellites were discovered about 1989. Were blood samples collected and stored?

Yes, blood samples were collected and stored.

L. 173-176: I'm confused here as well, can this be explained better?

We have modified this statement for clarity. Please see lines 175-179

Revised statement- *We tested the loci used in parentage assignments for deviations from Hardy-Weinberg equilibrium (HWE) and linkage disequilibrium using GenePop 4.7.5 [30] with 1000 dememorizations, 100 batches, and 1000 iterations per batch. We ran the parentage assignment analyses for 5-year time periods (1990, 1995, 2000, 2005, 2010, 2015) using 52–78 candidate parents in each year.*

L. 205-207: At this point it is not totally clear what happens when only one member of a co-breeder pair disappears. Is that considered a turnover?

No, the sole breeder of a particular sex in a group is not challenged for the position. Turnovers only occur when all breeders of a particular sex die or disappear. We have explained this in line 138.

Statement in manuscript: *Once established as a breeder, neither sex is challenged by larger same-sex coalitions – even as single breeders;*

L. 236: is 'nestling' here the same as 'fledgling' a few lines above?

Yes, we have changed nestling to fledgling to avoid confusion.

L. 271-273: 8.67 is more than 8.58, the results of some of these tests comparing 3-4 groups need to be better explained.

We have revised the result section considerably to make it easier to interpret. Please see lines 259-290. We have added this clarification to the results section to avoid confusion.

Most of P. 13: These results strongly suggest that there should be an analysis of the combined effects of number of cobreeders and territory quality.

In the revised manuscript, we have included an analysis looking at the effect of the number of cobreeders and territory quality as additive fixed effects on the lifetime direct fitness of both males and females. We conducted additional analyses of an interaction between territory quality and cobreeding, but felt that the idiosyncratic results distract from the main focus of the paper and thus have opted to not include them in this revision in an effort to keep the word count down and provide more clear messaging about our findings (please see results of model output below). Of course, if the Editor feels strongly that these analyses need to be included, we will be happy to oblige.

L. 285 : Extra “was”.

We have made this change

L. 367-379: Not sure I understand this - territory quality was determined by granary size, so what is the effect of a poor crop of acorns? More holes will not compensate for fewer acorns?

We agree that granary size is not an actual measure of resource availability. Acorn crops vary annually but territory quality is more consistent. A high quality territory, for example, is expected to remain a high quality territory across years whereas acorn crops will vary from year to year. It is the interaction of these two variables that has the largest influence on reproductive output. As you rightly point out, a large granary cannot compensate for a poor acorn crop. Likewise, a small granary cannot reap the benefits of a large acorn crop. Since acorns may be stored across years, previous research (Koenig et al. 2011, *American Naturalist* 178 145-158) has shown the significant effect of granary size in addition to acorn availability. Cobreeding is also more common

(especially among males) on territories with large granaries irrespective if they happen to be poorly filled in a poor acorn crop year.

Revised text: *Territory quality for acorn woodpeckers at Hastings is determined by the presence and size of granaries, specialised trees that are used to store acorns in individually drilled holes, sometimes for multiple years before they are consumed. Territory quality measured as granary size remains constant irrespective of annual fluctuation in acorn crop and the presence of breeding coalitions is determined primarily by the presence of granaries rather than the acorn crop in any particular year [17]. Granary size is therefore used as a proxy for territory quality (low-quality: < 2500 storage holes; high-quality >2500). All members of the social group participate in territory maintenance and defence [27].*

Fig. 1 panel A: The Y-axis label (frequency of cooperation) appears incorrect. Panel B: If there is a slash between offspring and breeding, I cannot see it.

We have made the recommended changes.

Marco Festa-Bianchet

Referee: 2

Comments to the Author(s)

This is a very interesting, well-written paper, which is remarkably error-free. It tackled the fitness costs and benefits of a rare social system where separate coalitions of males and females combine to breed in a polygynandrous association, and cooperate to rear a joint clutch. The data are based on an exemplary long-term study which is one of the best ever accumulated for a free-living population. The data are particularly remarkable because of features not mentioned in the paper, such as the difficulty in accessing nests high in trees.

The results are of great interest. While 'cooperation' appears to have several attendant costs, in the modal coalition for males (two) birds have higher lifetime reproductive success than birds that breed solitarily because they have longer reproductive spans and more breeding attempts. However, for females there are no benefits from being in a coalition. The authors hypothesise several reasons why this might be true

There are a few things that might have attracted discussion but are not addressed as much as I would like at the moment. First, while duos of males do best, the outcomes for larger coalitions are very poor (which are not infrequent). The authors present evidence that larger male coalitions may be better at taking over (and presumably defending) territories and its vital granary resource, which might explain a suboptimal group size, but I think this point requires further attention. Second, and following on from this, having a high quality territory dramatically increases success (Table S5). However,

the relevant models treat coalition size and territory quality as variable that do not interact. Although I generally admire parsimony in statistical analysis, in this case an interaction between coalition size and quality seems both biologically possible (eg Fig 2b) and potentially informative.

This is an interesting point. We conducted analyses to examine the apparent interaction between coalition size and territory quality. The results ended up being complex since we used Group ID as a random effect in the mixed-models. Each group has constant territory quality, and this, with territory quality also being included as a fixed effect produced small non-significant parameter estimates (see model output below). As they did not alter any of our main conclusions, we are inclined to keep them out of the manuscript in an effort to reduce the “noise” that tends to unnecessarily complicate the story. We would prefer to leave such analyses for a subsequent manuscript where we can focus more on this issue. Of course, we are happy to include them if the reviewer or editor feel that this is a deal breaker and that such analyses need to be included.

Males

Model: total young fledged ~ no. cobreeders * Territory Quality +(1 GROUP ID)			
	Estimate ± SE	Z	P
No. cobreeders	-0.42 ± 0.03	11.44	<0.001
Territory Quality	-0.54 ± 0.14	3.9	<0.001
No. cobreeders * Territory Quality	0.28 ± 0.15	1.8	0.09

Females

Model: total young fledged ~ no. cobreeders * Territory Quality +(1 GROUP ID)			
	Estimate ± SE	Z	P
No. cobreeders	-0.38 ± 0.05	6.66	<0.001
Territory Quality	-0.63 ± 0.15	4.02	<0.001
No. cobreeders * Territory Quality	0.40 ± 0.32	1.1	0.08

Thid, the explanation that large groups size facilitates territory acquisitions seems remarkably similar to arguments that have already been developed for cooperatively polyandrous societies such as the Galapagos hawk where only one female breeds, and this might be mentioned. Last, I have always thought that cooperative polygynandry and polyandry must be quite different, but in this case the difference seems to stem from the ability of both males and females in polygynandrous societies to achieve the optimum.

We do mention Galapagos hawks at line 80 in the context of polyandry.

Fig. 1 – Is it really true as suggested in the caption that the horizontal lines within the boxes represent the mean? It is conventional for the median to be depicted, and this is suggested by what appear to be identical values in some of the plots (eg C and D). Similarly, the whiskers often depict the 10 and 90th percentiles rather than quartiles. The points beyond the whiskers also imply this is also true.

We have revised the figure legend accordingly and thank you for catching this error.

Revised figure legend: **Figure 1:** *Individual fitness benefits as a function of the mean number of cobreeders (1= single breeder) in male and female acorn woodpeckers. Boxplots denote 10th and 90th percentile of direct fitness data with the horizontal line showing the median.....*

Appendix B

Dear Editor and Associate Editor,

Thank you for the constructive feedback we received on our Manuscript ID RSPB-2021-0579.R1 “Lifetime reproductive benefits of cooperative polygamy vary for males and females in the acorn woodpecker (*Melanerpes formicivorus*)”. We especially thank the Associate editor for highlighting some inconsistencies in the manuscript and feel that the changes they suggested have significantly improved the clarity of our findings. We have addressed the reviewers’ and Associate Editor’s comments in the main text and modified the figures and figure legends accordingly. We have also provided justification for the changes we made in the revised submission. We thank you again for giving us the opportunity to make these changes and hope that these revisions are satisfactory. The responses to comments are listed below and any changes made to the main text are in blue. See below for entire manuscript with changes highlighted in blue.

Sincerely,

Sahas Barve

Associate Editor

Board Member: 1

Comments to Author:

The authors have revised this manuscript, including additional analyses. One of the original reviewers looked through this revision, and had a few suggestions and queries remaining. Upon going through the revisions, I noted a few -mostly minor- additional issues that should be addressed in a revised manuscript and revised response to reviewer comments.

We thank the associate editor for these constructive comments which have made the manuscript easier to interpret.

First, it is not well argued why territory quality is quantified as two levels, which is easiest for the analysis of interactions. But these interactions are not included and results are described as if territory quality is continuous, e.g. in the supplemental tables the effect size of territory quality is given without reference category (i.e. implying it is a continuous variable) and at Question 6, “increasing territory quality”). This needs to be clarified, and can be solved - for example - by a short rationale at l. 148-149 why territory quality has two levels rather than continuous variation and slight tweaking of the results descriptions and clarifying in the supplemental tables output and legend how territory quality is defined.

Territory quality (measured as the size of the acorn granary; 2 levels, “low” and “high”) varies widely across the population, but largely remains the same for a particular group from year to year. Counting the exact number of granary holes is often impossible, since parts of the granary may not be fully visible or whether all holes are actually functional for storing acorns is difficult to discern. Our assessment of territory quality is thus based

on an estimated number of holes and it would be impractical to attempt to convert to a continuous variable. Territories can generally be broken down into groups that either have a small granary (up to 2500 holes; low quality) or groups that have large granaries (more than 2500 holes but these often range from 5000 to 10000+ holes; high quality). Newly founded groups or groups that have lost their granaries (e.g. burned in a fire, fell over in a storm) have few, if any, granary holes and are assigned a “low” territory quality, while groups that have been active for several decades and have accumulated granary holes over time, are given a “high” territory quality category. Territory quality measured in this way is only a gauge of how many acorns could be stored rather than a measure of how many acorns are actually stored. Hence, we feel confident that our classification of territory quality as either low or high is a sound ecological categorization as it affects acorn woodpecker life history (Barve et al. 2019, *American Naturalist* 190:830-840).

In the revised version, please see lines 146 -155 for a revision of the text in the Methods section describing why territory quality is not used as a continuous variable but a categorical one (see below).

“Territory quality, measured as granary size, remains relatively constant compared to annual fluctuations in acorn crops and breeding coalitions are often predicted primarily by the presence of granaries rather than the acorn crop in any particular year [17]. The exact number of functional holes in a group’s granary, which may be cryptically spread out over dead limbs in the canopies of several trees, is difficult to count accurately. Furthermore, while additional holes are added by group members on a more-or-less continuous basis, a major branch or entire granary may fall, episodically reducing granary size. Granary size is thus categorised as either low-quality: < 2500 storage holes; or high-quality >2500 (but usually more than 5000) holes. All members of the social group participate in territory maintenance and defence [27].

We have, thus, considered territory quality as a categorical variable (low or high). Following your directive, we have edited the description of Question 6 and included a reference category in our model output tables in the supplementary materials (tables S1, S3–S5). Please see revised supplementary tables, below, for changes.

Revised legends for Supplementary Table

Tables S1–S5: *In all models, single breeders were compared against other cobreeding units. In models S1, S3, S4 and S5 for the categorical variable “territory quality” (low or high), low-quality territory was used as the reference category and thus a positive estimate indicated that high territory quality had a positive effect on the response variable. For tables S6 and S7 please see table legend.*

Table S1. Effect of cobreeding / joint nesting on number of offspring per reproductive attempt. Generalised linear mixed model with number of offspring per brood as response variable.

Fixed effects:	Estimate	SE	Z	P
----------------	----------	----	---	---

Males				
(Intercept)	0.75	0.05	16.51	<0.001
Cobreeding duo	-0.27	0.06	-4.30	<0.001
Cobreeding trio	-0.37	0.08	-4.80	<0.001
4+ cobreeders	-0.53	0.10	-5.78	<0.001
Territory Quality (high)	0.10	0.06	-1.69	<0.001
Females				
(Intercept)	1.17	0.03	34.79	<0.001
Joint-nesting duo	-0.32	0.05	-5.91	<0.001
Joint-nesting trio	-0.55	0.10	-5.07	<0.001
Territory Quality (high)	-0.07	0.05	-1.32	0.185

Table S2. Effect of cobreeding / joint nesting on age at first breeding. Generalised linear mixed model with age at first breeding as response variable.

Fixed effects:	Estimate	SE	Z	P
Males				
(Intercept)	1.10	0.09	11.95	<0.001
Cobreeding duo	0.07	0.10	0.70	0.48
Cobreeding trio	-0.06	0.10	-0.62	0.53
4+ cobreeders	0.12	0.11	1.00	0.27
Females				
(Intercept)	1.05	0.07	13.80	<0.001
Joint-nesting duo	-0.01	0.08	-0.12	0.90
Joint-nesting trio	-0.16	0.15	-1.10	0.28

Table S3. Effect of cobreeding / joint nesting on reproductive lifespan. Generalised linear mixed model with years in group as breeder as response variable.

Fixed effects:	Estimate	SE	Z	P
Males				
(Intercept)	0.96	0.10	9.43	<0.001
Cobreeding duo	0.60	0.10	6.01	<0.001
Cobreeding trio	0.80	0.10	7.67	<0.001
4+ cobreeders	0.56	0.12	4.70	<0.001
Territory Quality (high)	0.14	0.06	2.06	<0.02
Females				
(Intercept)	1.38	0.08	16.27	<0.001
Joint-nesting duo	0.17	0.07	2.22	<0.03
Joint-nesting trio	-0.15	0.13	-1.15	0.24
Territory Quality (high)	0.15	0.08	-1.73	0.08

Table S4. Effect of cobreeding / joint nesting on number of lifetime reproductive attempts. Generalised linear mixed model with lifetime number of reproductive attempts as response variable.

Fixed effects:	Estimate	SE	Z	P
Males				
(Intercept)	0.51	0.12	4.03	<0.001
Cobreeding duo	0.78	0.12	6.54	<0.001
Cobreeding trio	1.01	0.12	8.11	<0.001
4+ cobreeders	0.87	0.13	6.20	<0.001
Territory Quality (high)	0.31	0.07	3.94	<0.001
Females				
(Intercept)	1.21	0.10	11.44	<0.001
Joint-nesting duo	0.14	0.08	1.61	0.10
Joint-nesting trio	-0.14	0.13	-1.05	0.29
Territory Quality (high)	-0.14	0.10	-1.36	0.17

Table S5. Effect of cobreeding / joint nesting on lifetime reproductive success. Generalised linear mixed model with lifetime number of young fledged as the response variable For territory quality, low-quality territory was used as the reference category and thus a positive estimate indicated that high territory quality had a positive effect on the response variable.

Fixed effects:	Estimate	SE	Z	P
Males				
(Intercept)	1.51	0.10	14.99	<0.001
Cobreeding duo	0.55	0.08	6.90	<0.001
Cobreeding trio	0.45	0.08	5.20	<0.001
4+ cobreeders	-0.01	0.10	-0.04	0.96
Territory Quality (high)	0.18	0.06	2.71	<0.007
Females				
(Intercept)	2.01	0.09	20.82	<0.001
Joint-nesting duo	-0.05	0.06	-0.96	0.33
Joint-nesting trio	-0.54	0.10	-5.36	<0.001
Territory Quality (high)	0.01	0.07	0.10	0.91

We decided to not include the interaction between territory quality and number of cobreeders because the results were not informative: territory quality remains largely unchanged for a particular group from year to year, and group ID was included as a random effect. This resulted in small and non-significant parameter estimates for interactions in our models (see model output below). As the interactions neither altered our main conclusions nor improved model fit, we are inclined to leave them out of the manuscript, although we are happy to report them in the supplement if the reviewers or editors feel it necessary. We would however remove non-significant interactions from the main manuscript to facilitate interpretation of the first-order effects.

Males

Model: total young fledged ~ no. cobreeders * Territory Quality +(1 GROUP ID)			
	Estimate ± SE	Z	P
No. cobreeders	-0.42 ± 0.03	11.44	<0.001
Territory Quality (high)	0.54 ± 0.14	3.9	<0.001
No. cobreeders *	-0.28 ± 0.15	1.8	0.09
Territory Quality (high)			

Females

Model: total young fledged ~ no. cobreeders * Territory Quality +(1 GROUP ID)			
	Estimate ± SE	Z	P
No. cobreeders	-0.38 ± 0.05	6.66	<0.001
Territory Quality (high)	0.63 ± 0.15	4.02	<0.001
No. cobreeders *	-0.40 ± 0.32	1.1	0.08
Territory Quality (high)			

Second, sample sizes have changed in several models, and outcomes have changed qualitatively and quantitatively (e.g. Supplementary Table questions 3, 4 and 5; table S3, reproductive lifespan, large differences for males and duo females). Why this occurred is however not addressed in the response to reviewers, so this should be briefly addressed in a revised response. Because of these additional changes, not all changes are tracked in the revised manuscript (response to reviewers document) and supplemental files. To gain a complete overview of changes, I request the authors re-submit a full comparison outlining all differences between the original manuscript (at first submission) and the final version (incorporating all previously made changes as well as all changes and suggestions requested here).

We apologize for not providing a more direct explanation for the change in sample sizes in the revised draft. The sample sizes changed because in this revised version we only included fitness data for adults assigned parentage through genetic parentage assignment methods (unlike the previous version where some adults were assigned parentage based on them being single breeders for parentage assignment). Hence, we analysed a different set of data for the revision. Our parentage data are now more robust and only include adults breeding between 1984 – 2016 (in the previous version the range was 1974 – 2016). These changes were described as such in Line 163 of the previously submitted revision and also highlighted in blue in other places throughout the manuscript submitted as part of the responses to reviewers document with the previous submission (as per the author guidelines). Changes in sample sizes are highlighted in lines 166, 262-271, 529-530 of the current submission. These changes to the data altered some of our results but the overall findings remain unchanged. However, we agree that should have highlighted these changes more explicitly in our previous version. We apologise and hope that these changes are now more clearly articulated.

We now include a version of the manuscript that highlights all of the changes we have made in both revisions from the original submission. Please see the manuscript with tracked changes below this letter for details.

Third, regarding the discussion of the new result that female duos have similar lifetime reproductive success to single females (in the previous manuscript version their lifetime reproductive success was lower): I am not sure I follow, if single females and duos have similar lifetime reproductive success, why this alone can explain cooperative polygamy in females (see l. 310-311; similar statement is made in Abstract), why it thus ‘reveals a potential driver of cobreeding in females’ (l. 363), and why you conclude ‘cobreeding duos seem to benefit in similar ways to males’ (l. 380-381). It might be good to expand this argument a little.

Thanks for this comment. We have revised the text in lines 312-316 and lines 362-366 and lines 381-386 of the revised manuscript respectively. See below

Lines 312–316

“Thus, for cobreeding male duos and trios, and female duos (the most common coalition sizes for those respective sexes outside of singletons), lifetime reproductive success alone may provide sufficient fitness benefits to explain the presence of cooperative polygamy, since lifetime reproductive success for male duos or trios is greater than or for female duos is equivalent to single breeders.”

Lines 362–366

“However, we found that cobreeding female duos had direct lifetime reproductive success comparable to single breeders. Given that female cobreeders are typically closely related, the total inclusive fitness benefits of cobreeding in duos are probably greater than previously assumed and thus both direct and indirect fitness may influence cobreeding by females.”

Lines 381–386

“For females, duos also benefit from cobreeding, with an increase in reproductive lifespan and an equivalent lifetime reproductive output compared to single breeding females despite the lack of an advantage attributed to higher territory quality. However, the higher physiological costs of cobreeding for females (e.g., egg destruction) may negate any potential benefits among cobreeding trios.”

Fourth, in addition to the query from the reviewer about Figure 1: the asterisks are confusing, it seem they can denote two different things? For example, Figure 1B the asterisks do not seem to concur with the results of model S1, since 2, 3 and 4 co-breeders are all different from N=1 (similar to Table S3, but panel Figure 1D shows 3 asterisks, not 1; likewise, in panel 1F the asterisk does not seem to concur with the output of model S5 –

Related to this, in l. 524-525: error in text as to what asterisks denote and text is confusing “...and asterisks denote model estimates where which are significantly

greater than for single breeders. Asterisk over a 'mean N breeder' category indicates it has model estimate that differed significantly from the reference category of single breeder”.

Thank you for pointing out this ambiguity. We have remade the figure to show statistical differences and the direction of those differences. The revised figure and figure legend is below.

Figure 1: Individual fitness benefits as a function of the mean number of breeders (1= single breeder) in male and female acorn woodpeckers (colours correspond to number of breeders). Boxplots denote 10th and 90th percentile of the data with the horizontal line showing the median. Raw data points are shown. **A)** Relative proportion of breeder composition per year by sex in the study population, across the 43 years of the continuous study. In any given year, roughly half the males and approximately 70% females were single breeders while the rest were in cobreeding coalitions; **B–F** show lifetime fitness data for birds from 1984–2006 (males N = 275, females N = 224). Figures show comparisons between single breeders versus cobreeding categories

(mean number of cobreeders throughout an individual's life). Pound signs (#) show groups with fitness values significantly ($P < 0.05$) lower than single breeders. Asterisks () shows groups with fitness values significantly higher than single breeders while 'n.s.' indicate groups with fitness values not significantly different than single breeders. **B)** Number of fledglings produced by a breeder per successful nesting attempt for the group; **C)** Age of first breeding (when first assigned parentage). **D)** Reproductive lifespan (number of years as a breeder in the population); **E)** Number of nesting attempts that reached the incubation stage by the group during the tenure of a breeder, and **F)** Lifetime number of young (assigned to an individual via genetic parentage analysis) that reached fledgeling stage.*

I. 313-314, word missing or redundancy here? “sex-specific differences in lifetime reproductive success between males and females“

Thank you for catching this redundancy. We have revised the statement. See below

“Previous research on this population suggests that these sex-specific differences in lifetime reproductive success are likely driven by at least two factors.....”

Reviewer(s)' Comments to Author:

Referee: 1

Comments to the Author(s)

I liked this paper when I first read it and I like it even better now. Excellent analysis, impressive sample size, reasonable interpretations.

We thank you for your comments and constructive feedback.

I have a few final suggestions:

L. 35: of nesting attempts (?).

Yes. Thank you for catching that error. See below for revised text.

“However, males nesting in duos and trios had longer reproductive lifespans, more lifetime nesting attempts, and higher lifetime reproductive success than those breeding alone”

L. 279: Either remove the 'either' or add 'or males in groups of 4 and larger'.

Done

Fig. 1: I do not understand A, partly because the axis label and the legend description do not match. Does the figure mean that about half of all males and 3/4 of females were single breeders?

We apologize for the lack of clarity in the figure legend. We aim to show that in any given year, about half the males and roughly 75% females are single breeders, while the other half are in cobreeding coalitions of varying sizes. We hope the revised figure legend and axis label are now clear.

Fig. 2: I am not totally sure I understand the natural history and some definitions here - presumably the 'initial' coalition size is the maximum during its history? If I understood correctly, established breeders are not challenged, so breeder turnover presumably is nearly always when the last bird dies, so that at that point there is one breeder left, even if earlier on there may have been 2 or more co-breeders?

Thank you for this comment. Yes, your interpretation is correct. However, we would like to point out that, 1) generally there is no correspondence between the number of breeders disappearing and the number of birds getting replaced and 2) in both males and females, it is not always the case that only one remaining breeder is replaced. In our dataset, there are several instances of multiple same-sex breeders disappearing together, for example during a particularly bad acorn crop year and such groups eventually only being re-occupied by a single breeder but also, and as you say, a single breeder being replaced by several cobreeders. However, for males on high quality territories, single breeders are generally replaced by a larger coalition, i.e. coalitions tend to be more successful at winning competitions for high quality territories, while such a pattern is not seen on low quality territories. In females, there is no relationship between territory quality and the replacing number of breeders.

The figures below show the number of turnovers for males and females relative to breeder turnover index. A turnover index of “-4” indicates 5 breeders being replaced by 1 breeder, while a turnover index of 6 means 1 breeder being replaced by 7 breeders.

Appendix C

Dear Editor and Associate Editor,

Thank you for the constructive feedback and for accepting our manuscript Manuscript ID RSPB-2021-0579.R2 “Lifetime reproductive benefits of cooperative polygamy vary for males and females in the acorn woodpecker (*Melanerpes formicivorus*)” for publication. The manuscript benefitted significantly from the feedback and review process and is much stronger with more clarity. As suggested by Dr. Neiman and the Associate Editor, we have added a line in the Results section of the revised manuscript indicating that the model outputs of the interaction term between number of cobreeders and territory quality is presented in the supplementary table (table S8) in this revised version.

See lines 290-291 for revised statement.

“For the results of a model investigating the interaction between number of cobreeders and territory quality, please see supplementary table 1.”

We are also adding the track changed version of the full manuscript below.

Thank you again.

Sincerely,

Sahas Barve